# Student-Oriented Teacher Knowledge Refinement for Knowledge Distillation

## ABSTRACT

Knowledge distillation has become widely recognized for its ability to transfer knowledge from a large teacher network to a compact and more streamlined student network. Traditional knowledge distillation methods primarily follow a teacher-oriented paradigm that imposes the task of learning the teacher's complex knowledge onto the student network. However, significant disparities in model capacity and architectural design hinder students' comprehension of the complex knowledge imparted by the teacher, resulting in sub-optimal learning results. This paper introduces a novel approach that emphasizes a student-oriented perspective and refining the teacher's knowledge to better align with the student's needs, thereby improving knowledge transfer effectiveness. Specifically, we present the Student-Oriented Knowledge Distillation (SoKD), which incorporates a learnable feature augmentation strategy during training to dynamically refine the teacher's knowledge of the student. Furthermore, we deploy the Distinctive Area Detection Module (DAM) to identify areas of mutual interest between the teacher and student, concentrating knowledge transfer within these critical areas to avoid spreading irrelevant information. This targeted approach ensures a more focused and effective knowledge distillation process. Our approach, functioning as a plug-in, could be integrated with various knowledge distillation methods. Extensive experimental results demonstrate the efficacy and generalizability of our method.

## CCS CONCEPTS

• **Computing methodologies → Computer vision**.

## KEYWORDS

Model Compression, Knowledge Distillation, Feature Augmentation

## 1 INTRODUCTION

Knowledge distillation (KD), first introduced by [16], has attracted significant interest in both academic and industrial research for its effectiveness in transferring knowledge from a pre-trained and high-performance teacher network into a more compact and lower-capacity student network. This knowledge transfer improves the student network's learning performance while preserving its structure. Knowledge distillation has been applied in various tasks such

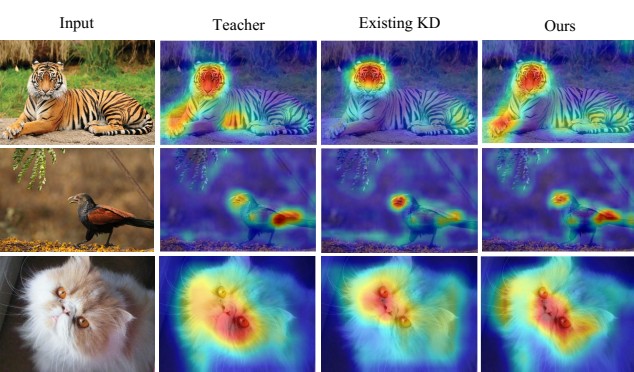

**Figure 1: This research is motivated by the observation that although a well-pretrained teacher network can precisely identify crucial regions within input data, its student counterpart, constrained by a smaller capacity and different architectural design, has difficulty understanding the teacher's recognition patterns. Our approach offers a refinement of the intricate teacher knowledge to cater to the student's needs, preserving the integrity of the overall teacher network knowledge. Ultimately, this enables students to identify the recognition patterns of the teacher network more accurately.**

as classification [18, 23], object detection [6, 50], and semantic segmentation [29, 40].

The original knowledge distillation [16] leverages soft labels provided by the teacher network to guide the student network. Subsequently, logits-based works have investigated various constraints through decoupled logits [52, 61] and more comprehensive constraints [13, 18]. However, since logits only provide information on the distribution at the class level and lack the comprehensive structural information of the input data, feature-based knowledge distillation [37] distills the student network through pixel-level constraints applied at the intermediate layers features have increasingly gained attention [7, 44, 56].

Figure 1 utilizes Grad-CAM [39] to visualize the crucial regions prioritized by the network, enabling an assessment of recognition patterns across different networks. The results indicate that due to substantial disparities in model capacity and architecture design, it is challenging for the student to fully assimilate the recognition patterns of the teacher from the intricate teacher knowledge, ultimately leading to sub-optimal learning outcomes. Existing approaches often facilitate the student's understanding of complex knowledge from the teacher via surrogate representation [26, 32, 45, 53, 56], or by implementing rigid constraint [15, 44, 51]. All these methods adopt a teacher-oriented perspective, assuming that the teacher's knowledge is fully applicable and beneficial to the students, neglecting the inherent differences in their capabilities and structural

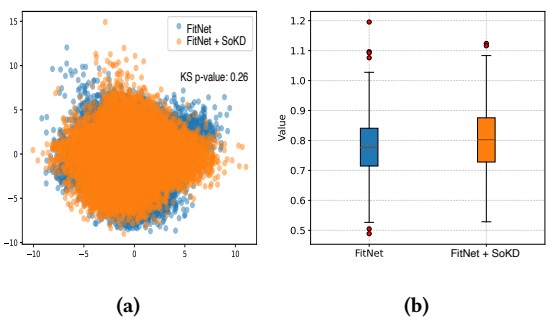

**Figure 2: Visualization and statistical analysis of features. (a): Visualization of features after dimensionality reduction through PCA. (b): Box plot of features. The experiment was conducted on a teacher-student pair of ResNet $32 \times 4$ and ResNet $8 \times 4$ on the CIFAR100 dataset.**

designs. Therefore, forcing students to learn complex knowledge from teachers often produces a sub-optimal result. Given this critical insight, we propose shifting to a student-oriented perspective that tailors teacher knowledge to the student's learning capabilities and architectural design. Therefore, a core issue in our method is: *how to appropriately adjust the teacher's knowledge within a reasonable scope to adapt to the needs of the student network.*

In light of this, data augmentation, known for diversifying input data through various transformations—emerges as a promising approach. Its capability to produce new data instances from the identical distribution [1] presents the possibility of refining the teacher's knowledge to align more closely with the student's learning requirements while striving to preserve the original distribution of the teacher's knowledge to prevent any destruction of the knowledge within the teacher. However, since enhancements at the input level are not directly related to the distilled knowledge (such as features from intermediate layers or logits from the final layer), the impact of data augmentation on input data for distillation remains uncontrollable. Furthermore, [4, 33] demonstrated that feature augmentation in high-dimensional spaces offers the advantage of increased plausibility of generated data points, thereby enhancing the likelihood of producing reasonable. Therefore, our strategy shifts towards leveraging the potential of augmentation at a finer level of granularity within the latent space. By implementing an augmenting strategy at latent space, we aim to directly tailor the teacher's knowledge, making it more accessible and relevant for the student network. However, the manually selected augmentation strategies not only require a significant amount of grid search time to find the optimal strategy, but they may also disrupt the distribution of the original teacher knowledge and cannot guarantee that the augmented features will be suitable for the student network. Inspired by neural network search [62], automatically searching for the optimal augmentation strategy provides a great idea. This automated search for feature-level augmentation strategies can avoid introducing human biases, prevent unreasonable augmentation strategies from undermining the original knowledge, and significantly reduce the time spent on grid searches for various augmentation strategies. Note that [24, 54] also uses augmentation for

knowledge distillation. However, their primary goal is to amplify the knowledge corresponding to the non-target categories in the label rather than achieving student-oriented knowledge adjustment. They expanded the teacher's knowledge, whereas this study seeks to tailor the teacher's knowledge to accommodate the students' requirements.

Based on the abovementioned analysis, the core idea of our proposed method is: *adjust the teacher's knowledge through a learnable feature augmentation strategy.* Specifically, we introduce Student-Oriented Knowledge Distillation (SoKD), an innovative perspective that dynamically tailors the pretrained teacher network's knowledge to the requirements of the student network. SoKD consists of two key components: Differentiable Automatic Feature Augmentation (DAFA) and the Distinctive Area Detection Module (DAM). DAFA is guided by student knowledge, searching for the most suitable augmentation strategy within a carefully designed feature augmentation search space. Furthermore, DAM utilizes shared parameters to identify areas of mutual interest between the teacher and student, facilitating knowledge transfer and easing the student's learning process. Figure 2a visualizes the results before and after enhancing the features, and also through the Kolmogorov-Smirnov test calculating the KS p-value (i.e., 0.26), we can find that the overall distribution remains unchanged after enhancement, indicating no destruction of the original teacher's knowledge. The box plot results in Figure 2b suggest that the enhancement notably increased the diversity of features while preserving the original scope of knowledge and significantly reducing outlier occurrences, simplifying the student's task of capturing the teacher's knowledge and minimizing the risk of misleading information. In summary, the main contributions of the paper are:

- From a student-oriented perspective, we proposed that SoKD adjusts teacher knowledge to accommodate the capacity and architectural design of the student network while preserving the overall integrity of the original teacher's knowledge.
- We apply DAFA to automatically learn the most suitable enhancement strategy for adjusting the teacher's knowledge through an automated search method and utilize DAM to identify mutual distillation areas, thereby improving information transfer efficiency and simplifying the student's learning process.
- SoKD can be plugged in existing knowledge distillation methods, and extensive experiments show that SoKD can significantly improve the performance of these methods.

## 2 RELATED WORK

### 2.1 Knowledge Distillation

Following the original work of knowledge distillation [16], a series of studies [18, 32, 60, 61] improves the representation of logits. These logits-based methods transfer the knowledge via minimizing the Kullback-Leibler divergence between the predicted logits of teachers and students. The feature-based distillation methods [37] use features from intermediate layers. They have garnered more attention, as the higher level logits-based methods lack structural information. However, a substantial gap between the teacher and student prevents the latter from fully acquiring the comprehensive knowledge of the former. Studies by [7, 13, 18, 23] promote

**Figure 3: The overall framework of SoKD, comprising two key components: 1) DAFA, a differentiable module for augmenting feature strategy search. This module adapts strategies during training, aiming to uncover knowledge that is more suitable for the student network. 2) DAM, which identifies distinctive areas between the teacher and student networks. This module focuses on areas of mutual interest for knowledge transfer, thereby avoiding unnecessary knowledge distillation.**

the student to learn knowledge as accurately as possible through more comprehensive and stringent constraints. In addition, many works [20, 32, 48] use a progressive distillation paradigm to avoid direct distillation when the gap between the teacher and student networks is large. Other methods [42, 44, 51] improve the transfer of knowledge to the student by refining the constraints. While [11, 30] recognized that teacher knowledge might not suit the student, they search student architectures adaptable to teacher knowledge from the student's perspective. However, searching for student network architectures is time-consuming and often yields architecture unfriendly to edge devices.

In summary, while existing knowledge distillation methods enhance student comprehension of teacher knowledge from student perspective, this paper adopts the teacher's perspective, tailoring its knowledge to accommodate the student.

## 2.2 Augmentation

In the past few years, handcrafted data augmentation techniques are widely used in training network. For example, rotation, translation, croping, resizing, and flipping are commonly used to augment training example. Beyond these, like Cutout [19], Mixup [58], and Cut-Mix [55]. Inspired by data augmentation [55, 58], current research boosts the network's representational ability by feature space augmentation. It is suggested that higher-level representations amplify the volume of credible data points in the feature space [3, 34]. Given that features are usually well linearized [46], it is therefore feasible to use simple vector interpolation [12] and mixing up [47]. Features are perturbed in the directions of intra-class/cross-domain variability [22], and instance features are directly synthesized by leveraging semantics [8]. Although these methods achieve promising improvements on the corresponding tasks, they need expert knowledge to design the operations and set the hyper-parameters for specific datasets. Recently, inspired by the neural architecture search (NAS) [62], some methods attempted to automate learning data augmentation polices. Our approach takes a student-oriented perspective and leverages automated strategy search at the feature level to dynamically adjust the teacher's knowledge to meet the student's needs.

## 3 METHODOLOGY

In this section, we will introduction our Student-Oriented Knowledge Distillation (SoKD). Our method has two core components: 1) Differentiable Automatic Feature Augmentation (DAFA) in Section 3.1, and 2) Distinctive Area Detection Module (DAM) in Section 3.2. The overall framework of SoKD is shown in Figure 3.

### 3.1 Differentiable Automatic Feature Augmentation

Our approach builds upon and improves the foundation of feature-based knowledge distillation. For a given set of inputs $x$, the general form of the feature-based knowledge distillation is:

$$\mathcal{L}_{\text{feat}} = (f^t(x) - g(f^s(x)))^2, \tag{1}$$

where $g(\cdot)$ is the mapping function transforming the student's feature to align with the teacher's feature, and $f^t$ and $f^s$ denote the teacher and student backbone blocks respectively. The total training objective for the student model is:

$$\mathcal{L}_{\text{train}} = \mathcal{L}_{\text{task}} + \gamma \mathcal{L}_{\text{feat}}, \tag{2}$$

where $\mathcal{L}_{\text{task}}$ is the standard task training loss for the student, and $\gamma$ is the corresponding weight.

Given that the parameters in the pre-trained $f^t$ are fixed, the teacher network is limited to providing knowledge with its own bias. The student may have difficulties to understand this complex and fine-grained knowledge, and this kind of knowledge itself may often be inappropriate for the student. In this study, we aim to

dynamically adjust the teacher network's knowledge $\mathcal{F}^t = f^t(x)$ to better suit the need of the student network.

To preserve teacher network knowledge without changing parameters, we favor feature-level over input augmentation considering the fact that higher-level representations expand the relative volume of plausible data points within the feature space [10]. To bypass manual enhancement's biases and time costs, we introduce DAFA, an NAS framework that dynamically tailors augmentation strategies to student needs during distillation.

*Feature Search Space.* We design a search space focused on feature representation, for simplicity and effectiveness. By analyzing existing state-of-the-art models, we develop a series of operations that can significantly enhance the robustness of feature representation, such as masking and adding noise.

In order to identify an enhancement strategy that meets the requirements within the minimum possible search time, we adopt a procedure inspired by Fast AutoAugment [25]. Given the knowledge $\mathcal{F}$ from the teacher, we wish to find a policy $s(\mathcal{F})$ which could adaptively adjust the teacher's knowledge during the training process, thereby meeting the learning needs of the student at the current stage.

Suppose the policy $s(\mathcal{F})$, denoted by $s$ for short, has $P$ sub-policies. Each sub-policy $s_i$, $1 \leq i \leq P$, has $k$ operations $O_j^{s_i}$ with the probability $p_j^{s_i}$ for $j = 1, \cdots, k$, or do not do any operation, i.e., keep $\mathcal{F}$ unchanged. Combing these two cases, each $s_i$ corresponds to $k$ operations

$$\bar{O}_j^{s_i}\left(\mathcal{F}; p_j^{s_i}, m_j^{s_i}\right) = \begin{cases} O_j^{s_i}\left(\mathcal{F}; m_j^{s_i}\right) & \text{with } p_j^{s_i}, \\ \mathcal{F} & \text{with } 1 - p_j^{s_i}, \end{cases} \quad (3)$$

for $j = 1, \cdots, k$, where $m_j^{s_i}$ is the magnitude of the operation. Thus, the complete sub-policy $s_i(\mathcal{F})$ can be represented by

$$s_i(\mathcal{F}) = \bar{O}_k \circ \bar{O}_{k-1} \circ \cdots \circ \bar{O}_1(\mathcal{F}), \quad (4)$$

for $i = 1, \cdots, P$.

*Feature Search Strategy.* After describing the operations in the feature search space, we now focus on the feature search strategy. Given that the selection of sub-policies is a discrete process, to facilitate the end-to-end training, we should make the search space continuous. Specially, the sub-policy selection and operations are sampled from Categorical and Bernoulli distributions, respectively. To select a specific sub-policy $s(\mathcal{F})$ and make the search space continuous, we relax the the categorical choice of a particular operation to a softmax

$$\bar{s}(\mathcal{F}) = \sum_{s \in S} \frac{\exp(\alpha_s)}{\sum_{s' \in S} \exp(\alpha_{s'})} s(\mathcal{F}) \quad (5)$$

over all possible operations, where $S$ is the set of all candidate sub-policies, and $\boldsymbol{\alpha} = (\alpha_1, \cdots, \alpha_{|S|})$ is a vector. At the end of search, a discrete feature augmentation strategy can be obtained with the most likely operation, i.e., $s(\mathcal{F}) = \arg\max_{s \in S} \alpha_s$.

Consequently, the task of searching for feature augmentation is simplified to learning a set of variables $\boldsymbol{\alpha}$ whose components are continuous.

After selecting a specific sub-policy using the above step, within this sub-policy we determine whether this operation is executed by

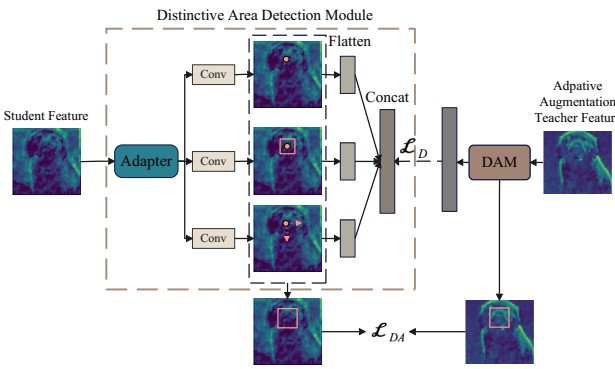

Figure 4: DAM in SoKD. Utilizing three head branches, DAM individually predicts the heatmap, size, and offset, thereby identifying the important areas of the feature. The teacher and student features are inputted into the corresponding DAM which with an identical structure and shared parameters, aiming at identifying distinctive areas that are of mutual interest to both the teacher and student networks.

sampling from a Bernoulli distribution. Essentially, this introduces a stochastic process, assigning a probability of execution or non-execution to each operation. The feature operation $\bar{O}$ with the application probability $\beta$ and magnitude $m$ can be represented as:

$$s(\mathcal{F}) = b \cdot O(\mathcal{F}; m) + (1 - b) \cdot \mathcal{F}, \quad b \sim \text{Bernoulli}(\beta). \quad (6)$$

After the relaxation process, the next step is to jointly optimize the feature augmentation strategy parameters $\gamma = \{\boldsymbol{\alpha}, \boldsymbol{\beta}, m\}$ and the student network weights $w$. We define $\mathcal{L}_{\text{train}}(w, \gamma)$ and $\mathcal{L}_{\text{val}}(w, \gamma)$ as the training and validation losses, respectively. The aim is to find $\gamma^*$ that minimizes the validation loss $\mathcal{L}_{\text{val}}$, with the optimal weights $w^*$ being derived by minimizing the training loss $w^* = \arg\min_w \mathcal{L}_{\text{train}}(w, \gamma^*)$:

$$\begin{aligned} \min_\gamma \quad & \mathcal{L}_{\text{val}}(w^*(\gamma), \gamma) \\ \text{s.t. } \quad & w^*(\gamma) = \arg\min_w \mathcal{L}_{\text{train}}(w, \gamma). \end{aligned} \quad (7)$$

To estimate the gradient of $\mathcal{L}_{\text{val}}$ with respect to parameters $\boldsymbol{\alpha}, \boldsymbol{\beta}, m$, the Gumbel-Softmax reparameterization trick is utilized to reparameterize the parameters $\boldsymbol{\alpha}, \beta$, making the gradient differentiable. With the Gumbel-Softmax reparameterization, Eq. (5) could be represented as:

$$\bar{s}(\mathcal{F}) = \sum_{s \in S} \frac{\exp\left((\log(\alpha_s) + g_s)/\tau\right)}{\sum_{s' \in S} \exp\left((\log(\alpha_{s'}) + g_{s'})/\tau\right)} s(\mathcal{F}), \quad (8)$$

where $g = -\log(-\log(u))$ with $u \sim \text{Uniform}(0, 1)$, and $\tau$ is the temperature of Softmax function.

Similarly, we apply the same reparameterization trick to the Bernoulli distribution

$$\begin{aligned} \text{Dis}(\lambda, \beta) &= \sigma\left(\left(\log\frac{\beta}{1 - \beta} + \log\frac{u}{1 - u}\right)/\lambda\right), \\ & u \sim \text{Uniform}(0, 1), \end{aligned} \quad (9)$$

such that the sigmoid function $\sigma$ is differentiable with respect to $\beta$.

**Table 1: Results of top-1 accuracy (%) for homogeneous architectures on CIFAR-100.**

| | Teacher | ResNet56 | ResNet110 | ResNet32×4 | WRN-40-2 | WRN-40-2 | VGG13 |
|---|---|---|---|---|---|---|---|
| | | 72.34 | 74.31 | 79.42 | 75.61 | 75.61 | 74.64 |
| | Student | ResNet20 | ResNet32 | ResNet8×4 | WRN-16-2 | WRN-40-1 | VGG8 |
| | | 69.06 | 71.14 | 72.50 | 73.26 | 71.98 | 70.36 |
| Logits | KD [16] | 70.66 | 73.08 | 73.33 | 74.92 | 73.54 | 72.98 |
| | DKD [61] | 71.97 | 74.11 | 76.32 | 76.24 | 74.81 | 74.68 |
| | MLKD [18] | 72.19 | 74.11 | 77.08 | 76.63 | 75.35 | 75.18 |
| | ND* [43] | 72.33 | 74.32 | 78.28 | 76.95 | 75.56 | 75.22 |
| Feature | FitNet [37] | 69.21 | 71.06 | 73.50 | 73.58 | 72.24 | 71.02 |
| | +DFKD | **71.54 (+2.33)** | **72.21 (+1.15)** | **74.41 (+0.91)** | **74.63 (+1.05)** | **73.02 (+0.78)** | **71.91 (+0.89)** |
| | CRD [44] | 71.16 | 73.48 | 75.51 | 75.48 | 74.14 | 73.94 |
| | +DFKD | **71.73 (+0.57)** | **73.81 (+0.33)** | **76.54 (+1.03)** | **76.54 (+1.06)** | **74.62 (+0.48)** | **74.37 (+0.43)** |
| | AT [56] | 70.55 | 72.31 | 73.44 | 74.08 | 72.77 | 71.43 |
| | +DFKD | **70.98 (+0.43)** | **72.93 (+0.62)** | **74.31 (+0.87)** | **75.15 (+1.07)** | **73.09 (+0.32)** | **71.64 (+0.21)** |
| | ReviewKD [7] | 71.89 | 73.89 | 75.63 | 76.12 | 75.09 | 74.84 |
| | +DFKD | **72.61 (+0.72)** | **74.63 (+0.74)** | **77.41 (+1.78)** | **77.02 (+0.90)** | **75.63 (+0.54)** | **75.31 (+0.47)** |

Since some operations in the search space are non-differentiable, we employ straight-through gradient estimator [2] to optimize the augmentation magnitude $m$. For a feature $\hat{\mathcal{F}} = s(\mathcal{F})$ augmented by sub-policy $s$, the influence of the augmentation operation on each pixel $(i, j)$ of the image is uniform, specifically, $\frac{\partial \hat{\mathcal{F}}_{i,j}}{\partial m} = 1$, the gradient of the magnitude can be calculated as:

$$\frac{\partial \mathcal{L}_{\text{val}}}{\partial m} = \sum_{i,j} \frac{\partial \mathcal{L}_{\text{val}}}{\partial \hat{\mathcal{F}}_{i,j}} \frac{\partial \hat{\mathcal{F}}_{i,j}}{\partial m} = \sum_{i,j} \frac{\partial \mathcal{L}_{\text{val}}}{\partial \hat{\mathcal{F}}_{i,j}}. \quad (10)$$

Through the aforementioned reparameterization trick, we have transformed the non-differentiable feature search into a differentiable operation, hence making it possible to optimize parameters through gradient updates.

### 3.2 Distinctive Area Detection Module

DAM is the second module of SoKD. As shown in the right part of Figure 3, both the student's feature and the teacher's feature after DAFA are fed into DAM, and DAM outputs the distinctive areas. The purpose of DAM is to address the challenge that even when the bias in the knowledge provided by the teacher is alleviated by DAFA, fully replicating the teacher's comprehensive information remains difficult.

DAM aims to decouple the feature, thereby enabling the separation of distinctive areas for the transmission of knowledge in these distinctive areas. Specifically, we first employ an adapter to facilitate the mapping of the student and teacher features into a common semantic space. Following this, we use a multi-branch detection head to pinpoint distinctive areas. The DAM module is composed of three branches, each consisting of consecutive convolutional layers, i.e., $3 \times 3$, and $1 \times 1$. The output of DAM is:

$$A = \mathcal{D}(\text{conv}_i(\Phi_s(\mathcal{F}^s))), \quad (11)$$

where $\text{conv}_i$ ($i = 1, 2, 3$) represents the three branches of the DAM module, $\Phi_s$ aligns the input between the teacher and the student,

and $\mathcal{D}$ stands for the decode part of DAM, which is used to generate distinctive areas. During the training process, the features of both the student and teacher are passed through a shared-parameter DAM module to predict the distinctive areas independently. The predicted results are then supervised using $L_2$ loss, facilitating the alignment of distinctive areas between the student and teacher. The final training loss for the DAM module is:

$$\mathcal{L}_D = \left(\text{conv}_i(\Phi_s(\mathcal{F}^s)) - \text{conv}_i(\bar{s}(\mathcal{F}^t))\right)^2. \quad (12)$$

After distinctive areas are filtered using DAM, Eq. 1 is modified to the following representation:

$$\mathcal{L}_{\text{DA}}\left(\mathcal{F}^s, \mathcal{F}^t\right) = \sum_{i=1}^{N} \left(\mathcal{M}(A_i)\Phi_s(\mathcal{F}^s) - \mathcal{M}(A_i)\bar{s}(\mathcal{F}^t)\right)^2, \quad (13)$$

where $N$ is the number of the distinctive areas, and $\mathcal{M}$ is a mask operation that generates the corresponding mask based on the distinctive areas $A_i$.

### 3.3 Objectives for Optimization

For $\mathcal{L}_{\text{val}}$ in Eq. (7), to ensure that the enhanced feature provides suitable knowledge for the student, we use a consistency loss to make the features as close as possible to $\mathcal{F}^s$ after applying the corresponding sub-policy $s$. Therefore, the searched strategy can adjust the teacher's knowledge to fit the student network:

$$\mathcal{L}_{\text{con}} = \frac{1}{2}(s(f^t(x)) - f^s(x))^2. \quad (14)$$

For the distillation of the student network, in addition to the loss related to the original task and the distillation loss at the feature level, we also carry out more coherent knowledge distillation based on DAM in Sec. 3.2. Thus, the final loss function can be expressed as:

$$\begin{aligned} \min_{\gamma} \quad & \mathcal{L}_{\text{con}}\left(w^*(\gamma), \gamma\right) \\ \text{s.t.} \quad & w^*(\gamma) = \arg\min_{w}\left(\mathcal{L}_{\text{task}} + \alpha\mathcal{L}_D + \beta\mathcal{L}_{\text{DA}}\right), \end{aligned} \quad (15)$$

where $\alpha$, $\beta$ represent corresponding weights.

Table 2: Results of top-1 accuracy (%) for heterogeneous architectures on CIFAR-100.

| | | ResNet32×4 79.42 | WRN-40-2 75.61 | VGG13 74.64 | ResNet50 79.34 | ResNet32×4 79.42 |
|---|---|---|---|---|---|---|
| | Teacher | | | | | |
| | Student | ShuffleNetV1 70.50 | ShuffleNetV1 70.50 | MobileNetV2 64.60 | MobileNetV2 64.60 | ShuffleNetV2 71.82 |
| Logits | KD [16] | 74.07 | 74.83 | 67.37 | 67.35 | 74.45 |
| | DKD [61] | 76.45 | 76.70 | 69.71 | 70.35 | 77.07 |
| | MLKD [18] | 77.18 | 77.44 | 70.57 | 71.04 | 78.44 |
| | ND* [43] | 77.01 | 77.25 | 70.94 | 71.19 | 78.76 |
| Features | FitNet [37] | 73.59 | 73.73 | 64.14 | 63.16 | 73.54 |
| | +DFKD | **74.93 (+1.34)** | **75.65 (+1.92)** | **66.32 (+2.18)** | **67.12 (+3.96)** | **74.21 (+0.67)** |
| | AT [56] | 71.73 | 73.32 | 69.40 | 68.58 | 72.73 |
| | +DFKD | **73.24 (+1.51)** | **75.09 (+1.77)** | **69.64 (+0.24)** | **68.75 (+0.17)** | **73.43 (+0.70)** |
| | ReviewKD [7] | 77.45 | 77.14 | 70.37 | 69.89 | 77.78 |
| | +DFKD | **78.12 (+0.67)** | **77.32 (+0.18)** | **70.79 (+0.42)** | **71.10 (+1.21)** | **78.64 (+0.86)** |
| | CRD [44] | 75.11 | 76.05 | 69.73 | 69.11 | 75.65 |
| | +DFKD | **75.73 (+0.62)** | **77.29 (+1.24)** | **70.44 (+0.71)** | **69.57 (+0.46)** | **76.24 (+0.59)** |

Table 3: Top-1 and Top-5 accuracy (%) of student networks on ImageNet validation set.

| | Teacher | Student | AT | +DFKD | CRD | +DFKD | ReviewKD | +DFKD |
|---|---|---|---|---|---|---|---|---|
| | | | ResNet34 as the teacher, ResNet18 as the student | | | | | |
| top-1 | 73.31 | 69.75 | 70.69 | **72.13 (+1.44)** | 71.17 | **71.86 (+0.69)** | 71.61 | **72.53 (+0.92)** |
| top-5 | 91.42 | 89.07 | 90.01 | **91.34 (+1.33)** | 90.13 | **90.71 (+0.58)** | 90.51 | **91.32 (+0.81)** |
| | | | ResNet50 as the teacher, MobileNetV1 as the student | | | | | |
| top-1 | 76.16 | 68.87 | 69.56 | **69.83 (+0.27)** | 71.37 | **71.60 (+0.23)** | 72.56 | **73.02 (+0.48)** |
| top-5 | 92.86 | 88.76 | 89.33 | **89.58 (+0.25)** | 90.41 | **90.69 (+0.28)** | 91.00 | **91.22 (+0.22)** |

Through the optimization of the aforementioned bi-level problem, we can determine the optimal feature augmentation strategy, thereby optimizing the student network under the proposed distillation framework.

## 4 EXPERIMENTS

In this section, we first provide a detailed introduction to the implementations of our experiments. Subsequently, we conduct comparisons with mainstream methods on various datasets and tasks. We also provide an analysis for further insights.

### 4.1 Experimental Settings

*Baselines.* We conducted extensive comparative experiments on teacher-student pairs across various neural network architectures [14, 17, 31, 38, 41, 57, 59] to validate the effectiveness of our method.

Our method can be integrated as a plug-in technique with various feature-based knowledge distillation approaches to enhance their performance. We applied our SoKD to existing distillation frameworks including FitNet [37], CRD [44], AT [56] and ReviewKD [7].

*Datasets.* We employed three prominent datasets to evaluate our methodologies. The first dataset is the CIFAR-100 [21], which

includes 60,000 images in 100 unique classes. Each image is 32x32 pixels in resolution. The dataset is partitioned into two sections: a training subset with 50,000 images and a test subset with 10,000 images. The second dataset is the ImageNet [9], an essential dataset for benchmarking in image classification. It contains approximately 1.3 million training images and 50,000 validation images, spread across 1,000 different classes. The ImageNet dataset is notable for its high-resolution images. Finally, the MS-COCO dataset [28] was also utilized, predominantly used for object detection tasks. This dataset includes images categorized into 80 different classes, with a training set of 118,000 images and a validation set of 5,000 images.

*Implementation Details.* For experiments on CIFAR-100, we adhere to the basic settings of the original experiments [7, 37, 61]. We set the batch size to 64, the learning rate to 0.05, and use the SGD [5] optimizer. The training is conducted on NVIDIA-A100 for 240 epochs. For ImageNet, we adopt the same training strategy as previous methods, with a batch size of 512, an initial learning rate of 0.1, and a total of 100 training epochs. Furthermore, the learning rate is decreased at epochs 30, 60 and 90. For COCO, we use the object detection framework of Detector2 [49] and conduct comparative experiments. All codes are implemented based on PyTorch [35].

**Table 4: Object detection results on MS-COCO. We take Faster-RCNN with FPN as the backbones.**

| | AP | AP50 | AP | AP50 | AP | AP50 |
|---|---|---|---|---|---|---|
| | ResNet101 & ResNet18 | | ResNet101 & ResNet50 | | ResNet50 & MobileNetV2 | |
| Teacher | 42.04 | 62.48 | 42.04 | 62.48 | 40.22 | 61.02 |
| Student | 33.26 | 53.61 | 37.93 | 58.84 | 29.47 | 48.87 |
| FitNet | 34.13 | 54.16 | 38.76 | 59.62 | 30.2 | 49.8 |
| +DFKD | **35.09 (+0.96)** | **54.93 (+0.77)** | **39.43 (+0.67)** | **60.08 (+0.46)** | **31.43 (+1.23)** | **50.85 (+1.05)** |
| FGFI | 35.44 | 55.51 | 39.44 | 60.27 | 31.16 | 50.68 |
| +DFKD | **36.32 (+0.88)** | **56.32 (+0.81)** | **39.78 (+0.34)** | **60.64 (+0.37)** | **32.02 (+0.86)** | **51.23 (+0.55)** |
| ReviewKD | 36.75 | 56.72 | 40.36 | 60.97 | 33.71 | 53.15 |
| +DFKD | **37.21 (+0.46)** | **57.52 (+0.80)** | **40.43 (+0.07)** | **61.86 (+0.89)** | **34.24 (+0.53)** | **54.29 (+1.14)** |

## 4.2 Main Results

*CIFAR-100.* To fully demonstrate the efficacy of our method, we conducted extensive comparative experiments with various teacher-student pairs on CIFAR-100. Tables 1 and 2 present the experimental results for homogeneous and heterogeneous architectures, respectively. The results indicate that SoKD significantly enhances the performance of original knowledge distillation in structurally similar teacher-student pairs (e.g., an improvement of 2.33 percentage points on FitNet for the ResNet56-ResNet20 pair). Moreover, it effectively transfers knowledge from the teacher to the student even in pairs with larger structural differences (e.g., an improvement of 3.96 percentage points on FitNet for the ResNet50-MobileNetV2 pair).

*ImageNet.* The results shown in Table 3 illustrate that SoKD can still achieve satisfactory performance on the more challenging dataset ImageNet. When employing ResNet34 as the teacher and ResNet18 as the student, out method improves the top-1 accuracy of AT from 70.69% to 72.13%. Notably, SoKD also significantly improves performance over the current state-of-the-art ReviewKD, increasing top-1 accuracy from 71.61% to 72.53%.

*Object Detection.* We extend our experiments to object detection, another fundamental computer vision task. Using Faster-RCNN [36]-FPN [27] as the backbones and adopting average precision (AP), AP50 as evaluation metrics, the results in Table 4 indicate a comprehensive enhancement of existing distillation methods through the combination with SoKD. This also effectively demonstrates the generalizability of out method.

## 4.3 Ablation Study

SoKD primarily comprises two crucial modules: DAFA and DAM. Additionally, we enhance the training process of DAFA through a consistency loss $\mathcal{L}_{con}$. We validate the effect of each component using a one-by-one approach, and conduct experiments with ResNet $32 \times 4$-ResNet $8 \times 4$ and VGG13-MobileNetV2, using ReviewKD as the baseline. The results in Table 5 demonstrate that each module within SoKD exhibits significant effectiveness.

**Table 5: Ablation for different modules in DFKD. R-324 and R-84 respectively denote ResNet $32 \times 4$ and ResNet $8 \times 4$.**

| Teacher & Student | DAFA | $\mathcal{L}_{con}$ | DAM | Accuracy (%) |
|---|---|---|---|---|
| R-324 & R-84 | - | - | - | 75.63 |
| | ✓ | ✗ | ✗ | 76.85 |
| | ✓ | ✓ | ✗ | 77.01 |
| | ✓ | ✓ | ✓ | **77.41** |
| VGG13 & MV2 | - | - | - | 70.37 |
| | ✓ | ✗ | ✗ | 70.52 |
| | ✓ | ✓ | ✗ | 70.67 |
| | ✓ | ✓ | ✓ | **70.79** |

**Table 6: Influence of search epoch number in DAFA.**

| Search Epochs | R110 & R32 | VGG13 & MV2 |
|---|---|---|
| 10 | 74.21 | 70.43 |
| 20 | **74.63** | **70.79** |
| 25 | 74.49 | 70.71 |
| 30 | 74.52 | 70.59 |

## 4.4 More Analysis

*Search epochs.* Since excessive feature augmentation may cause potential undermine of original knowledge, we also identify the optimal balance. This balance is crucial for effectively enhancing the knowledge provided by the teacher model, thereby offering more robust guidance for the student model. Table 6 presents the comparative results between different numbers of search epochs for two sets of teacher-student pairs on CIFAR-100: ResNet110-ResNet32 and VGG13-MobileNetV2. The results indicate that we actually need only a few epochs for searching to determine the most suitable feature augmentation strategy to adjust the knowledge in the teacher network. Strategies searched with more epochs are not only time-consuming but also prone to overfitting.

*Comparison of Manually Designed Augmentation Strategies.* To fully demonstrate the superiority of our approach, we compared it with manually designed augmentation strategies, including direct data augmentation on the input and several simple combinations

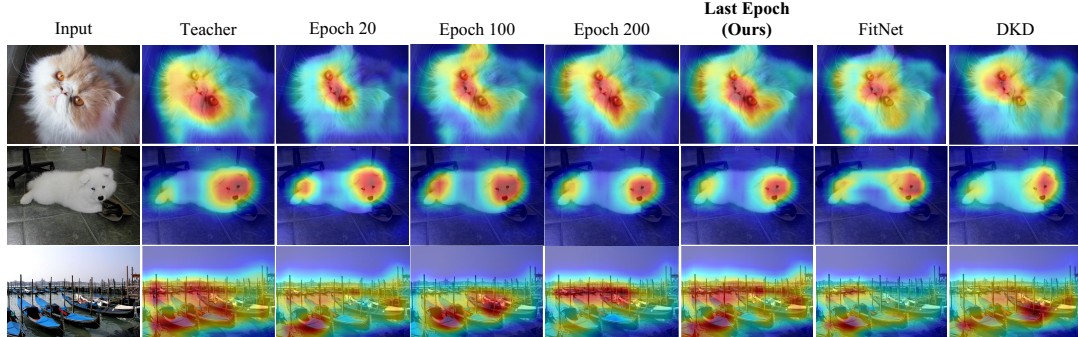

**Figure 5: In the distillation process on ImageNet, with ResNet34 serving as the teacher and ResNet18 as the student, the evolution of crucial regions within features. The final results are compared against FitNet and DKD.**

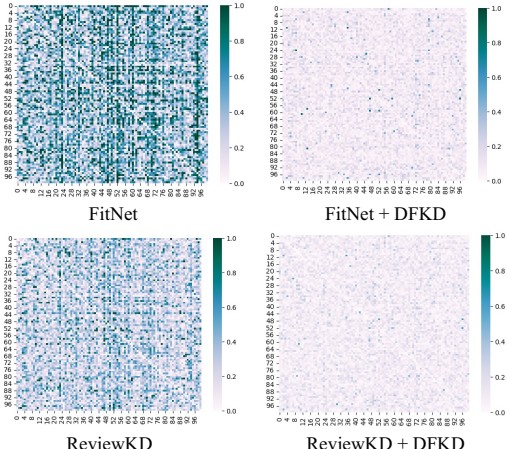

**Figure 6: Difference of student and teacher outputs. SoKD leads to a significantly smaller difference than baseline.**

**Table 7: Experimental comparison results of different types of augmentations based on FitNet for various teacher-student pairs.**

| Method | ResNet56
ResNet20 | ResNet32x4
ResNet8x4 | WRN-40-2
ShuffleNetV1 |
|---|---|---|---|
| Data Aug | 67.50 | 72.25 | 69.27 |
| Feature Aug | 69.72 | 73.28 | 72.52 |
| Aug Search (Ours) | **71.54** | **74.41** | **75.65** |

of augmentation strategies on features (e.g., adding noise, masking, and channel shuffling). The results in Table 7 indicate that augmentation on features performs better than direct augmentation on input data, which aligns with our expectations. Additionally, the results show that mere augmentation does not yield significant improvements. The core of our method lies in adjusting the teacher's knowledge based on the student's actual needs rather than relying solely on augmentation. The significant performance enhancement achieved by our method underscores its rationality and effectiveness.

*Visualization.* Figure 5 uses Grad-CAM visualizations to show how the student network's focus areas change during the training process for ResNet34-ResNet18 distillation on ImageNet. The results show that the student gradually learns the teacher's recognition patterns. Compared to other methods, SoKD achieves a closer recognition pattern to the teacher.

Figure 6 compares the final logits of the teacher and student models, highlighting differences. Previous feature-based methods, focusing only on intermediate layer features, often led to notable logit disparities due to latent space complexity. This disparity made imitating the teacher challenging for the student. However, with SoKD adjusting the teacher's intermediate knowledge, the student better understands and aligns this knowledge, achieving similar logit outputs crucial for the final task.

## 5 CONCLUSION

In this paper, we argue that the current teacher-oriented knowledge distillation often imposes the challenging task of learning complex teacher knowledge on the student network, frequently leading to sub-optimal outcomes. Therefore, we introduce a novel student-oriented knowledge distillation approach that employs automatically searched feature augmentation strategies. Without undermining the original knowledge of the teacher, this method appropriately adjusts the teacher's knowledge to accommodate the student network's model capacity and architectural design requirements. As a plug-in, SoKD significantly improves the performance of existing knowledge distillation methods on various datasets.

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
