# OpenReview forum: "Student-Oriented Teacher Knowledge Refinement for Knowledge Distillation"
_acmmm.org/ACMMM/2024/Conference — MM2024 Poster_

### Official Review · Reviewer_45D2 · 2024-05-16

**Rating:** 4
**Confidence:** 1

**Summary:**

The paper introduces a novel approach called Student-Oriented Knowledge Distillation (SoKD) that shifts from the traditional teacher-oriented paradigm to a student-oriented perspective. The main goal is to refine the teacher's knowledge to better align with the student's needs, improving knowledge transfer effectiveness. SoKD incorporates a learnable feature augmentation strategy and a Distinctive Area Detection Module (DAM) to identify critical areas of mutual interest between the teacher and student, focusing knowledge transfer within these areas. The approach aims to enhance the student's learning process by tailoring the teacher's knowledge to the student's capacity and architectural design. Experimental results demonstrate the efficacy and generalizability of the method.

**Strengths:**

1. SoKD significantly enhances the performance of original knowledge distillation in teacher-student pairs with similar structures, and effectively transfers knowledge from teachers to students even in pairs with large structural differences.

2. SoKD is still able to achieve satisfactory performance on the more challenging ImageNet datasets, significantly improving the performance of the current state-of-the-art methods.

3. By combining with SoKD, the existing distillation method has been comprehensively enhanced in the target detection task, demonstrating the generalization ability of the method.

**Limitations:**

1. Just as the author mentioned, this paper is not the first to propose Student-Oriented optimization methods for models. Therefore, compared to related approaches based on similar principles, the distinctive contributions of this paper have not been adequately expressed.

2. In the experimental part, the selection and adjustment process of some key parameters is not described in detail enough, and there is a lack of specific guidance on experimental reproduction, which may affect the reader's confidence in the experimental results.

**Suitability:**

2

---

### Official Review · Reviewer_ifdB · 2024-05-24

**Rating:** 3
**Confidence:** 3

**Summary:**

The paper presents student-oriented knowledge distillation, termed SoKD, where a learnable feature augmentation strategy was developed to dynamically refine the teacher’s knowledge of the student so that it is adapted to the student’s learning capacity. Furthermore, a distinctive area detection module is proposed to focus the distillation on the areas of mutual interest between student and teacher. The experimental results show that baseline KD models with the proposed SoKD perform better than their counterparts without SoKD.

**Strengths:**

* The idea of refining the teacher’s knowledge from the student's perspective is interesting. It is not unknown that the learning capacity discrepancy between teachers and students in KD is a big challenge, attracting a significant amount of attention in the KD community.
    * To deal with the discrepancy and bridge the gap, the paper proposes a feature augmentation method to a NAS-based feature augmentation method that dynamically tailors augmentation strategies to student needs during distillation. The use of Gumbel-softmax makes the feature augmentation process differentiable.
    * Aligning distinctive areas between the teacher and students can force the KD to focus on those aligned areas, resulting in another KD loss function
    * Experimental work shows that existing feature-based models can benefit from the proposed method.

**Limitations:**

* The performance gain given by the proposed method is somewhat marginal, particularly compared with some logic-based KD methods, as shown in the tables. It is recommended that the authors either report the standard errors or carry out statistical tests.
    * The presentation of this paper can be further improved, particularly the writing. The current manuscript has many English issues, like grammar. For example,
        * Line 157: “enhancing the likelihood of producing reasonable” it looks like the sentence is incomplete.
        * Line 167: “neural network search ” or  ”neural architecture search”?
        * Line 321: introduction -> introduce

**Suitability:**

3

---

### Official Review · Reviewer_b8Ao · 2024-05-27

**Rating:** 4
**Confidence:** 2

**Summary:**

This paper proposes to refine the teacher network knowledge from the student-oriented perspective for knowledge distillation. To achieve this, an automatic policy learning strategy is devised to find a suitable  sub-policy for the teacher network's feature augmentation, by which the teacher's knowledge is refined and adapted  to the student's capacity.
Besides, a Distinctive Area Detection Module is also employed to align the learning of student with the teacher network.
The motivation of this work is innovative for me, at least, and the proposed approach is also intuitive.
The experiments also validate the effectiveness of the proposed approach as well as some modules and strategies.

**Strengths:**

1) introducing a new research perspective for knowledge distillation, i.e., refining the teacher's knowledge to adapt the student network;
2) The paper is well-organized, easy to follow the proposed work;
3) Executing extensive experiments to validate the proposed approach.

**Limitations:**

1) the motivation of this work is creative， but the introducing of the motivation is too tortuous to catch in the third to the fifth paragraphs of Section 1,  especially for the fourth paragraph.  The audience may get trapped in the lengthy descriptions of why leverage auto augment for knowledge distillation.
2) The proposed distinctive area detection module seems to be very common, which seems not to provide many contributions to the student-oriented teacher refinement framework. I mean, similar techniques or ideas have been widely used in other tasks, which are not unique to the proposed framework. Despite its provided performance improvement, the novelty of this module seems to be limited.
3) There exist some unconvincing points in the experiments.
 Firstly, as shown in table 1, the performance of the proposed approach is not better than logist-based approaches, but the authors argued in Section 1 that logist-based approaches are not effective, achieving  sub-optimal learning. How to demonstrate the argumentation?
Secondly, except for tables 1 and 2, all subsequent experiments are under different settings, i.e., the teacher and student networks are different, especially for tables 5 to 7, which is very strange. Therefore, the experiments of this paper need to be suspected.
Finally, for NAS-based approaches, the model/algorithm runtime is an important metric. Besides, the model complexity regarding model parameters, latency, and  FLOPs  should be also reported. However, the above metrics in experiments are missing.

**Suitability:**

3

---

### Meta-Review · Area_Chair_FTnA · 2024-07-01

**Recommendation:** Accept (Poster)
**Confidence:** 4

**Metareview:**

The paper introduces a novel approach named Student-Oriented Knowledge Distillation. The main goal is to refine the teacher's knowledge to better align with the student's needs, improving knowledge transfer effectiveness. SoKD incorporates a learnable feature augmentation strategy and a Distinctive Area Detection Module (DAM) to identify critical areas of mutual interest between the teacher and student. The approach aims to enhance the student's learning process by tailoring the teacher's knowledge to the student's capacity and architectural design. Experimental results demonstrate the efficacy and generalizability of the method.

After the rebuttal, the paper receives 1 weak accept, 1 borderline accept, and 1 borderline reject. The AC consider the authors well solved the ifdB's concern in the rebuttal. Reviewer ifdB didn't respond after the authors' rebuttal.